# COVID-19 Booster Dose Coverage and Hesitancy among Older Adults in an Urban Slum and Resettlement Colony in Delhi, India

**DOI:** 10.3390/vaccines11071177

**Published:** 2023-06-29

**Authors:** Nandini Sharma, Saurav Basu, Heena Lalwani, Shivani Rao, Mansi Malik, Sandeep Garg, Rahul Shrivastava, Mongjam Meghachandra Singh

**Affiliations:** 1Department of Community Medicine, Maulana Azad Medical College, New Delhi 110002, India; nandini.sharma@nic.in (N.S.); shivani.rao@delhi.gov.in (S.R.); mm.singh22@gov.in (M.M.S.); 2Indian Institute of Public Health—Delhi, Public Health Foundation of India, New Delhi 122002, India; saurav.basu@iiphd.org (S.B.); mansidr676@gmail.com (M.M.); 3Department of Internal Medicine, Maulana Azad Medical College, New Delhi 110002, India; sandeep.garg@gov.in; 4Department of Biotechnology, National Biopharma Mission, Biotechnology Industry Research Assistance Council (BIRAC), New Delhi 110003, India; user-039@birac.nic.in

**Keywords:** COVID-19, SARS-CoV-2, vaccine hesitancy, booster dose, booster dose hesitancy, urban health, variants of concern

## Abstract

Background: The high prevalence of vaccine booster hesitancy, with the concomitant waning of humoral vaccine or hybrid immunity, and the emergence of SARS-CoV-2 variants of concern can accentuate COVID-19 morbidity and mortality. The study objective was to ascertain the COVID-19 vaccination coverage, including the administration of precaution (booster) dose vaccination, among the older population in an urban slum and resettlement colony population in Delhi, India. Methods: We conducted a cross-sectional survey in an urban resettlement colony, slum, and village cluster in the Northeast district of Delhi among residents aged ≥50 years. Results: A total of 2217 adults (58.28%) had obtained a COVID-19 booster (precaution) dose vaccine, 1404 (36.91%) had received two doses of a COVID-19 vaccine without booster dose, 121 (3.18%) were unvaccinated, while 62 (1.63%) participants received a single dose. Based on adjusted analysis, older adults (>65 years), higher education, and higher per-capita income were statistically significant predictors of booster dose vaccination. Conclusions: More than four in ten adults in an urban slum and resettlement colony in Delhi lacked COVID-19 booster dose vaccination despite high rates of double-dose vaccination (~95%). Public health programming should provide an enhanced focus on reducing complacency with renewed prioritization for improving ease of access to COVID-19 vaccination services, particularly in underserved areas.

## 1. Introduction

The World Health Organization has reported more than 44 million confirmed cases of COVID-19 in India [1]. Protection from COVID-19 infection and disease was initially dependent on social isolation and hygiene practices and afterwards predominantly focused on COVID-19 vaccination [2]. However, the protection conferred by the primary vaccination series wanes over time, especially against variants of concern linked to the SARS-CoV-2 Omicron virus, identified as having increased transmissibility, with diminished effectiveness of existing public health containment measures [3].

Booster COVID-19 vaccination was recommended globally to increase the durability of the humoral immune response, preserve vaccine effectiveness, and prevent recurrent infections, hospitalization, and severe diseases, especially against the Omicron variant possessing immune escape properties [4,5]. Several studies have reported that booster mRNA vaccine doses promote immunogenicity and increase the COVID-19 vaccine efficacy [6,7,8]. A systematic review of 51 studies suggested that booster mRNA vaccine shots conferred a small additional protective effect against Omicron variants [3]. Moreover, booster mRNA vaccine doses had no newer safety concerns as the majority of the adverse effects were minor immune system reactions to the booster vaccine dose [9]. Furthermore, both homologous and heterologous mRNA COVID-19 booster vaccine doses demonstrated satisfactory immunogenicity, acceptable safety, and effectiveness [10].

In India, the primary COVID-19 vaccination campaign was mostly driven using two WHO-approved vaccines: Covishield (ChAdOx1 nCoV-19), licensed from Oxford AstraZeneca (AZD222), a recombinant adenoviral vector vaccine [11] and Covaxin (BBV152), a whole-virion inactivated vaccine [12]. In January 2022, a homologous booster dose with the nomenclature “precaution dose” was launched for those who had completed the two-dose primary COVID-19 vaccination with a gap of 39 weeks between the first and second doses. Initially, the eligibility was limited to frontline workers, comorbid individuals, and older adults aged 60 and over. The booster dose eligibility was extended to all adults from April 2022 onwards throughout the country [13]. Both Covishield and Covaxin demonstrated acceptable levels of efficacy in preventing severe hospitalization and mortality and safety against serious adverse effects in real-world settings [14,15,16]. Furthermore, booster doses of both these vaccines have shown immunogenicity and neutralizing capacity against Omicron-related variants of concern (B.1.1.529) [17,18].

Vaccine hesitancy as per the WHO signifies the “delay in acceptance or refusal of vaccines despite availability of vaccine services” [19]. In Delhi, India, nearly universal single-dose COVID-19 vaccine coverage in the adult population was achieved by December 2021 [20]. Vaccine hesitancy for booster doses of COVID-19 vaccines can be driven by factors, such as complacency due to diminishing morbidity and mortality from Omicron-derived variants of SARS-CoV-2, perceived safety due to a history of previous infection expected to improve antibody titers, and the lack of confidence in vaccines from fear of side effects [21,22]. Conversely, the perceived effectiveness of booster doses against severe COVID-19 illness and inhibition of community transmission is observed as a significant predictor of their acceptance, especially in European health worker populations [23].

The high prevalence of booster dose vaccine hesitancy, with the concomitant waning of humoral vaccine or hybrid immunity, and the emergence of SARS-CoV-2 variants of concern through mutations conferring novel immune escape mechanisms are likely to prolong and perpetuate the pandemic with increased but avoidable COVID-19-related morbidity, mortality, and economic costs [21,22,23,24,25,26].

Vaccine hesitancy, especially among older citizens, continues to be a significant obstacle to the governments’ efforts to execute COVID-19 immunization initiatives in several countries [27,28]. A survey across 23 countries highlighted that one in eight responders who have received COVID-19 vaccination (12.1%) was unwilling to receive COVID-19 booster vaccine shots [29]. A large observational study conducted in the United Kingdom found that initial unwillingness in taking primary series of COVID-19 vaccines was also likely to translate into a higher likelihood of hesitancy for COVID-19 booster vaccine doses [30].

Combating vaccine hesitancy can be achieved through evidence-based interventions, including the rapid development and deployment of COVID-19 vaccine boosters (VB). These interventions are high-priority health system interventions towards reducing the further negative impact of the COVID-19 pandemic [22]. However, COVID-19 booster vaccination coverage in India, despite the free-of-cost provision by the government, is unsatisfactory during the Omicron wave of the pandemic [31,32]. Moreover, a steady and significant increase in the COVID-19 case burden associated with the circulation of the highly transmissible XBB1.16 variant is being reported since March 2023 [33].

Older adults and the elderly with a higher likelihood of weaker immune systems and underlying illnesses have several folds higher risk of COVID-19-associated mortality, especially in the absence of vaccination [34,35]. Consequently, it becomes imperative to ascertain the coverage of COVID-19 booster doses among vulnerable populations, such as older adults, especially those living amidst adverse social determinants as in urban slums. As per the WHO, urban slums usually lack one or more of the basic amenities, including access to improved water and sanitation, availability of sufficient living area, durability of housing, and security of tenure [36].

In this regard, people living in urban slums, especially in South Asia, constitute a highly vulnerable population due to pre-existing higher vaccine hesitancy and a lack of vaccine confidence secondary to perceived difficulty in vaccine accessibility [37,38]. Furthermore, as per the World Bank estimates, almost 49% of the urban population in India is residing in urban slums [39]; while in Delhi, the nation’s capital, it is estimated that nearly 34% of the population lives in slum settlements and 12.7% in slum resettlement areas [40]. However, there is a paucity of information on COVID-19 booster dose coverage rates and their determinants among populations living in urban slums in India.

We therefore conducted this study with the objective to ascertain the COVID-19 vaccination coverage, including booster (precaution) dose vaccination, among the older population in an urban slum and slum resettlement colony population in Delhi, India. Furthermore, we also assessed the willingness to pay for COVID-19 booster doses in the study participants.

## 2. Methods

Design, site, and participants: We conducted a cross-sectional survey as part of a larger study for the development of a Demographic Developmental and Environmental Surveillance Site (DDESS), focusing on an urban resettlement colony, slum, and village cluster in the Northeast district of Delhi, which is estimated to have a total population of 50,000. The site was purposively selected since the population in this area experiences adverse social determinants, including low income and lower educational standards. Furthermore, the site is an ongoing DDESS with geocoding and mapping of all households in the area, including data collection on the demographic information of all the residing members, with the attainment of high levels of community engagement that facilitates rapid surveys [41]. The data were collected from December 2022 to March 2023.

The participants included people over the age of 50 years who were residents of the study area for at least six months. The primary outcome of the study was the proportion of participants who had received the booster dose of any SARS-CoV-2 vaccine. The secondary outcome was the proportion of participants who had received two doses of any SARS-CoV-2 vaccine. Booster dose vaccine hesitancy was defined as the absence of vaccination with any authorized homologous or heterologous COVID-19 vaccine available in India.

Sample size and sampling: The study population of the area, which met the eligibility criteria of being aged ≥50 years, was ~7000. This information was obtained from the census conducted as part of the development of the DDESS. The sample size of 3824 was estimated at 95% confidence intervals, expecting 25% SARS-CoV-2 booster coverage, absolute precision of 2%, with a design effect of 2, and 10% inaccessible (locked) or non-responsive households. As part of the sampling strategy, the area was divided into 16 numbered clusters, 5 of which were purposively selected including the slum, village cluster, and adjoining slum resettlement blocks, followed by a universal sampling of all households having at least one eligible member, within each of the selected clusters consecutively. The line list of households having older resident members was obtained from the geocoded census data of the area, previously collected while implementing the DDESS project. Within a selected household having more than one eligible member, all the consenting eligible members were recruited in the study. If a household was locked, a second visit was planned seven days afterwards to reassess the status. Overall, 2530 households were approached by the field staff, of which 2353 households were successfully accessed. Within these households, 3804 eligible participants were recruited for the study (Figure 1).

Data were collected electronically using EpiCollect5 (Centre for Genomic Pathogen Surveillance, 2023), an Android tablet application, by trained field investigators. Information was collected from the participants through face-to-face interviews in the local language, Hindi. The interviews focused on the participants’ COVID-19 vaccination status (the vaccine name, doses, and the date of the last vaccination) which was also verified with their paper or electronic vaccine certification records. Additionally, information on the sociodemographic and relevant clinical characteristics of the participants was also obtained. Reasons for the COVID-19 precaution (booster) dose vaccine hesitancy and vaccine acceptability were also queried from the participants who had not received the booster dose and those who had already received the booster dose, respectively. A questionnaire adapted from a previous state-level cross-sectional survey in Delhi was used to ascertain the COVID-19 booster dose-associated vaccine hesitancy and acceptability among the participants [20].

Statistical Analysis: The data were analyzed using Stata (StataCorp. 2017. Stata Statistical Software: Release 15. StataCorp LLC, College Station, TX, USA). Descriptive statistics of the sociodemographic, clinical, and vaccination status parameters of the participants were reported. Based on bivariate analysis, the association between participant characteristics (independent variable) and both primary and booster dose vaccination status (outcome variable) was assessed using the chi-square test. The age of the participants was categorized into two categories: >65 (elderly) and those aged 50–65 years (older adults), as the severity of SARS-CoV-2 reinfection was expected to be greater in the former age group. For each independent variable, the respective unadjusted odds ratio (OR) with a 95% confidence interval (CI) was calculated. Assumptions of logistic regression, including linearity of independent variables with log odds of the outcome and multicollinearity, were assessed. The significance level was set at 0.05, and variables with a *p*-value less than 0.05 in the unadjusted analysis were included in the final adjusted logistic regression model. Furthermore, a forward step-wise selection method was adopted to construct the final regression model, starting from a null model until all the significant variables from the crude analysis were included. Model diagnostics such as outliers were checked in the final adjusted model. The results of the adjusted analysis were expressed as adjusted odds ratios (aORs) with their corresponding 95% CI. Separate analyses were performed for each outcome variable. Finally, the goodness of fit of the regression models was evaluated using the Hosmer–Lemeshow test, where a *p*-value greater than 0.05 indicated a good fit of the model.

Ethical considerations: The study was approved by the Institutional Ethics Committee, Maulana Azad Medical College and Associated Hospitals, New Delhi, vide F.1/IEC/MAMC/86/04/2021/NoH52 dated 28 September 2021 with an amendment for collection of data on COVID-19 vaccination questions. Written and informed consent was obtained from all the study participants for the collection of the sociodemographic and morbidity data. Additionally, verbal and electronic consent was obtained specifically for vaccination-related questions. The non-vaccinated and non-boosted participants were informed of the significance of COVID-19 vaccination, irrespective of their history of prior SARS-CoV-2 infection, and recommended to get themselves vaccinated at the nearest health facility.

## 3. Results

We recruited 3804 participants, including 1964 (51.63%) males and 1840 (48.37%) females. Most of the participants (*n* = 3208, 87.10%) were vaccinated with the Covishield vaccine, while 12.44% (*n* = 458) were administered the Covaxin vaccine. A total of 2217 adults (58.28%) had received the booster (precaution) dose of a COVID-19 vaccine apart from two primary series doses, while 1404 (36.91%) participants had received two doses of a COVID-19 vaccine but not the booster dose. Moreover, 121 participants (3.18%) were unvaccinated, while 62 (1.63%) participants had received a single dose of any COVID-19 vaccine.

The sociodemographic distribution of the study participants categorized by the number of COVID-19 doses administered to them is reported in Table 1. The 50–65 age group represented the majority (59.47%) of those who received two doses of the COVID-19 vaccine. Among those with a history of incomplete vaccination (either single dose or non-vaccination), a majority (56.83%) belonged to the older adult (>65 years) subgroup. Furthermore, nearly half (50.47%) of those who received the booster shot were over 65 years. A substantial proportion of illiterate adults (41.53%), comorbid adults (51.91%), and people with low per-capita income under Rs. 4000 (53.55%) did not receive even two doses of a COVID-19 vaccine. Nearly four in five (81.78%) of those who took the booster dose reported 2022 as the year of the last COVID-19 dose received.

On bivariate analysis, the participants aged >65 years (OR = 1.36 (1.19, 1.54), *p* < 0.001), having higher educational levels (OR = 1.76 (1.48, 2.09), *p* < 0.001), those suffering from comorbid conditions (OR = 1.20 (1.05, 1.37), *p* = 0.008), and with higher per-capita income (OR = 1.44 (1.09, 1.90), *p* = 0.01) had significantly higher odds of receiving the booster dose of a COVID-19 vaccine. Furthermore, based on adjusted analysis, older adults aged >65 years (aOR = 1.58 (1.37, 1.83), *p* < 0.001), higher education (aOR = 1.77 (1.41, 2.22), *p* < 0.001 for graduate and above), greater per-capita income (aOR = 1.46 (1.07, 2.00), *p* = 0.01 for >4000 rupees), and more recent year of last vaccine dose (aOR = 2.58 (1.89, 3.53), *p* < 0.001 for the year 2022) were significant predictors of booster vaccination (Table 2). Additionally, individuals receiving Covishield (aOR = 1.33 (1.08, 1.64), *p* < 0.001) compared to Covaxin were more likely to have availed the COVID-19 booster dose.

Among those who did not take the COVID-19 booster vaccine dose, a majority of respondents (50.84%) reported other reasons as their justification for not availing booster shot which included responses as follows: “did not have time”, “scared of pain from third dose”, “other health issues”, etc. The reasons for not availing the COVID-19 booster vaccine dose included perception that the pandemic was over (22.96%), unavailability of a booster dose in their near government facility (21.65%), perception of low risk of infection (18.14%), unaware of the availability of a booster dose (15.18%), previously experiencing vaccine side effects (7.89%), and belief that vaccines were ineffective in preventing the disease (3.77%).

On the contrary, among individuals who took the COVID-19 booster vaccine dose, most (88.76%) reported their behavior being influenced by advice from community health workers, 64.4% believed that the booster dose protected against serious COVID-19 disease, 61.88% reported being positively influenced by mass media channels, 41.66% took the booster dose on the advice of a family member, and only 3.05% took the booster dose based on the advice of a physician.

Those who had not received the booster vaccine dose were asked if they would be willing to pay for the booster vaccine dose. More than three in four non-boosted participants (77.63%) reported that they were unwilling to pay for the booster dose and that it should be provided without charge. The proportion of non-boosted participants who were unwilling to pay for booster dose was greater in participants with lower per-capita income (<4000 rupees) (51.54%), having no education (36.62%), and in those having no comorbid conditions (72.24%).

Unwillingness to pay for a booster dose in non-boosted participants was lower in those with lower per-capita income (<4000 rupees) (51.44%), illiterate (36.62%), and having no comorbid conditions (72.24%). In contrast, non-boosted participants having a positive willingness to pay for booster dose was observed to be higher in the participants aged 50–65 years (55.14%) and those with comparatively higher per-income (>4000 rupees) (62.31%). On adjusted analysis, the odds of lack of willingness to pay for a booster dose of COVID-19 vaccine were significantly higher among individuals who received the last dose of vaccine in 2022 (aOR = 1.52 (1.16, 1.98)) whereas significantly lower among participants having comparatively higher per-capita income (OR = 0.71 (0.53, 0.95)), higher educational level (aOR = 0.49 (0.33, 0.72)), and those suffering from one or more comorbid conditions (aOR = 0.73 (0.55, 0.97)) (Table 3).

## 4. Discussion

This study conducted in a socioeconomically disadvantaged urban setting observed that two doses of a COVID-19 vaccine were administered to 95.19% of participants while the proportion of unvaccinated participants was very low (3.18%). The booster vaccine coverage rates nearly one year after their introduction in our study settings (58.28%) were comparable with the intention for booster dose vaccination in US adults (61.8%) prior to their regulatory approval [27]. However, the booster hesitancy rates in our study (33.4%) are higher than that among Chinese adults (17.2%) [28] and German university staff (12.2%) [22]. A previous web-based survey in India in the second quarter of 2022 reported significantly lower (28.4%) coverage of the booster dose of COVID-19 vaccines [42]. Another study from India reported 44.1% booster dose hesitancy early during its introduction although a convenience sample was recruited which severely limited the generalizability of the study findings [43].

In our study, 87% of the participants had received the Covishield vaccine and were also more likely to have received the booster dose, while studies from Europe reported a rapid decline in the public demand for AZD1222 (Oxford-AstraZeneca) vaccine due to reports of increased thrombotic events that triggered massive vaccine hesitancy against the vaccine [22]. The finding of greater trust in the Covishield vaccine is probably correlated with the low incidence of serious adverse effects following immunization (AEFIs) in the country [44].

The findings of the present study enable the development of a COVID-19 booster dose conceptual framework mediated by the classical health belief model indicative of higher acceptance of the booster dose in older adults (aged 65 years) who are targeted by mass-media IEC campaigns for COVID-19 booster dose vaccination, have comparatively higher education, improved SES, and had recently completed the primary series of COVID-19 vaccination (Figure 2).

Similar to previous studies, higher educational status was linked to lower rates of booster vaccine hesitancy in this study suggestive of utilization of mass media (IEC) that increased rates of vaccination [22,28]. Similarly, a lack of perceived risk of serious illness reduced booster vaccine acceptability; however, unlike some other studies, the prior history of COVID-19 infection was not associated with reduced odds of booster vaccination [22,28]. Nevertheless, in contrast to a study in China, older people had higher odds of being vaccinated with a booster dose in this study [28]. Furthermore, in our study, advise from community health workers was found to be a key factor driving booster vaccination, corroborating global evidence of their trustworthiness in local communities as credible sources of information, thereby enabling the promotion of COVID-19 vaccination and inhibiting vaccine hesitancy [45,46].

In our study, a small but significant proportion (~5%) of participants were unvaccinated or only partially unvaccinated with a single dose of COVID-19 vaccine. Furthermore, the perceived lack of continued availability of COVID-19 vaccines in their nearby local health facility was reported as a barrier to vaccination by the non-boosted participants. As unvaccinated individuals continue to be at higher risk of serious COVID-19 disease, hospitalization, and death [47,48], facilitating vaccination of non-vaccinated and partially vaccinated individuals through reinvigoration of door-to-door campaigns that were highly effective in increasing vaccination coverage previously warrants prioritization [20]. Moreover, completion of the primary course of COVID-19 vaccination accords strong protective effectiveness against serious diseases from most extant variants of concern [3].

Although authorization for newer vaccines to provide alternatives to encourage heterologous boosting has been introduced in India [49], our findings indicate that most respondents, especially those belonging to lower socioeconomic status, were unwilling to pay for COVID-19 booster vaccine doses, thereby necessitating the ongoing need for the government to remain the principal provider of vaccine-related services. The willingness to pay for COVID-19 vaccines was significantly higher in a study conducted in Indonesia (~78%), which is probably explained by the differential standard of living among the populations surveyed [50].

The present study has certain strengths. First, the large sample size of this study ensured narrow confidence intervals of the outcome variables indicative of the high validity of the findings. Second, the study was conducted among the highly vulnerable population, including older adults and elderly people, with low socioeconomic status living amidst adverse-health-related social determinants in overcrowded, dense unplanned urban neighborhoods with a high risk of transmission of respiratory infections. Third, data were collected through face-to-face interviews with validation of vaccination status with vaccine certificates in most of the participants. The study limitations were that, in real-world settings, interviews were of brief duration, and hence awareness (health literacy) and attitudes towards COVID-19 vaccination which is a key determinant of vaccine confidence were not explored in-depth through qualitative assessment [51]. Nevertheless, our findings suggest that reasons for booster vaccine hesitancy were primarily driven by complacency (the COVID-19 pandemic was over), convenience (perceived unavailability of booster vaccination services in nearest health facilities), and confidence (perception of low risk of severe disease) [19]. Another limitation is the study was conducted in a single site in a relatively low-income, low-educational neighborhood, and hence, the findings cannot be generalized to divergent demographic and sociocultural settings.

In conclusion, more than four in ten adults in an urban slum and resettlement colony in Delhi lacked COVID-19 booster dose vaccination, despite high rates of double-dose vaccination (~95%), with comparatively fewer older adults (50–65 years), lower education levels, and lower per-capita income being predictors of booster dose hesitancy. Public health programming should provide an enhanced focus on improving COVID-19-vaccine-related health literacy, particularly to reduce complacency against emergent variants of concern with high disease potential, with renewed prioritization of enhanced accessibility to COVID-19 vaccination services in underserved areas.

## Figures and Tables

**Figure 1 vaccines-11-01177-f001:**
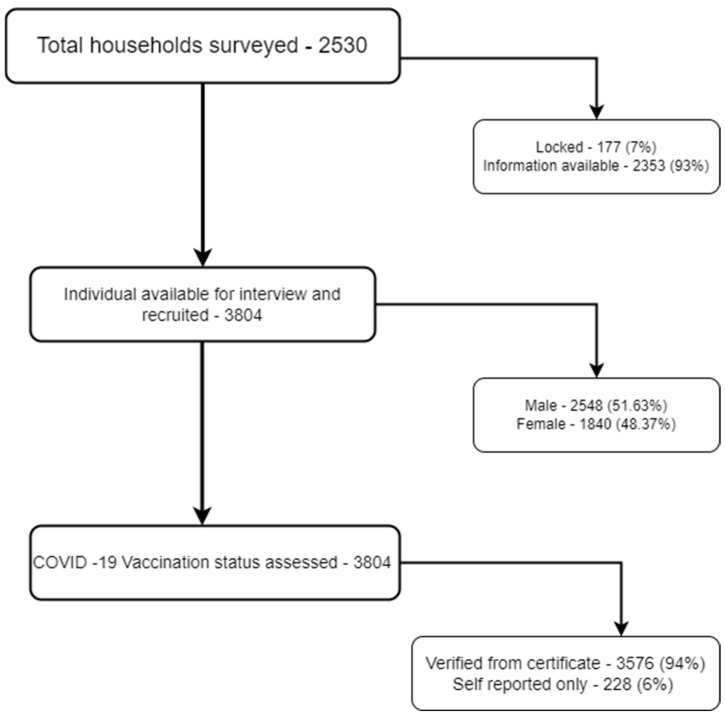
Recruitment of the participants.

**Figure 2 vaccines-11-01177-f002:**
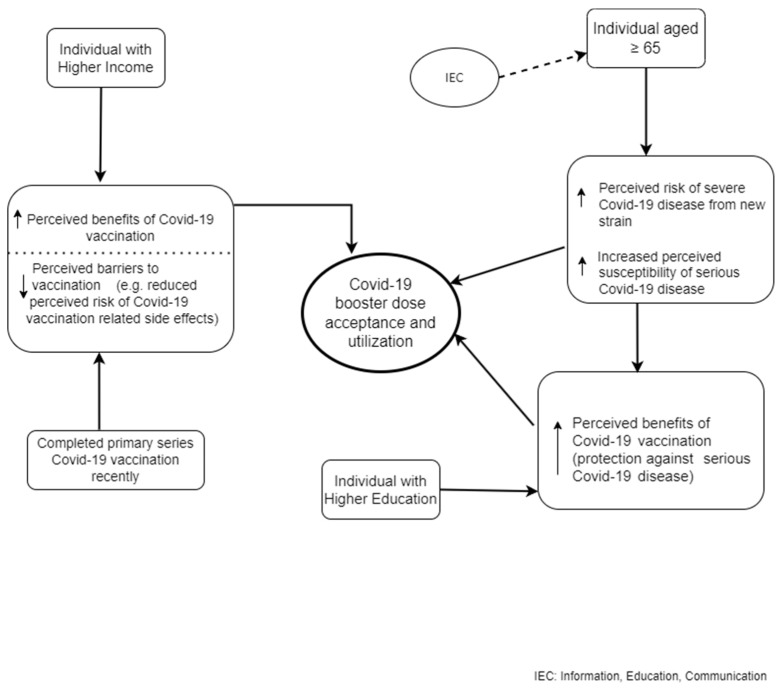
Conceptual framework of COVID-19 booster dose vaccine acceptance.

**Table 1 vaccines-11-01177-t001:** Sociodemographic characteristics of the participants stratified by the number of COVID-19 doses taken.

Variables	Incomplete Dose ^c^ (*n* = 183)	Two Doses without Booster (*n* = 1404)	Booster Taken (*n* = 2217)	Total (*n* = 3804)
Age (in years)				
50–65	79 (43.17)	835 (59.47)	1098 (49.53)	2012 (52.89)
≥66	104 (56.83)	569 (40.53)	1119 (50.47)	1792 (47.11)
Sex				
Male	92 (50.27)	697 (49.64)	1175 (53.00)	1964 (51.63)
Female	91 (49.73)	707 (50.36)	1042 (47.00)	1840 (48.37)
Education				
Illiterate	76 (41.53)	485 (34.54)	618 (27.88)	1179 (30.99)
Primary	39 (21.31)	270 (19.23)	374 (16.87)	683 (17.95)
Secondary/High school	36 (19.67)	340 (24.22)	546 (24.63)	922 (24.24)
Inter certificate/Graduate and above	32 (17.49)	309 (22.01)	679 (30.63)	1020 (26.81)
PCI ^a^ (in Rupees)				
≤4000	98 (53.55)	693 (43.36)	900 (40.60)	1691 (44.45)
>4000	85 (46.45)	711 (50.64)	1317 (59.40)	2113 (55.55)
Healthcare worker				
No	182 (99.45)	1386 (98.72)	2178 (98.24)	3746 (98.48)
Yes	1 (0.55)	18 (1.28)	39 (1.76)	58 (1.52)
Comorbidities				
None	88 (48.09)	957 (68.36)	1372 (61.89)	2417 (63.61)
≥1 Disease present	95 (51.91)	443 (31.64)	845 (38.11)	1383 (36.39)
Type of COVID-19 vaccine				
Covaxin	9 (14.52)	193 (13.75)	256 (11.55)	458 (12.44)
Covishield	51 (82.26)	1199 (85.40)	1958 (88.32)	3208 (87.10)
Other	2 (3.23)	12 (0.85)	3 (0.14)	17 (0.46)
COVID-19 infection (confirmed)				
Never	176 (96.17)	1336 (95.16)	2073 (93.50)	3585 (94.24)
Only once	7 (3.83)	51 (3.63)	113 (5.10)	171 (4.50)
More than once	0 (0.00)	17 (1.21)	31 (1.40)	48 (1.26)
COVID-19-related oxygen/hospitalization requirement				
No	182 (99.45)	1379 (98.22)	2176 (98.15)	3737 (98.24)
Yes ^b^	1 (0.55)	25 (1.78)	41 (1.85)	67 (1.76)
Year of last COVID-19 vaccine dose				
2021	39 (62.90)	559 (39.81)	404 (18.22)	1002 (27.21)
2022	23 (37.10)	845 (60.19)	1813 (81.78)	2681 (72.79)

Legend: ^a^ Per-capita income (PCI) in rupees. ^b^ Includes both me and my family member. ^c^ Includes “no dose” and “only one dose”.

**Table 2 vaccines-11-01177-t002:** Distribution of the factors associated with the utilization of COVID-19 booster vaccine dose (those who have taken it versus those who have not).

Variables	Crude OR (95% CI)	*p*-Value	aOR (95% CI)	*p*-Value
Age (in years)				
50–65	Ref		Ref	
>65	1.36 (1.19, 1.54)	<0.001	1.58 (1.37, 1.83)	<0.001
Sex				
Female	Ref		Ref	
Male	1.12 (0.98, 1.28)	0.07	0.92 (0.78, 1.07)	0.28
Education				
Illiterate	Ref		Ref	
Primary	1.07 (0.88, 1.29)		1.07 (0.87, 1.32)	
Secondary/High school	1.27 (1.07, 1.51)		1.33 (1.09, 1.64)	
Inter certificate/Graduate and above	1.76 (1.48, 2.09)	<0.001	1.77 (1.41, 2.22)	<0.001
PCI ^a^ (in Rupees)				
≤4000	Ref		Ref	
>4000	1.41 (1.23, 1.60)	<0.001	1.31 (1.12, 1.52)	<0.001
Healthcare worker				
No	Ref		Ref	
Yes	1.36 (0.78, 2.35)	0.26	0.94 (0.52, 1.68)	0.84
Comorbidities				
None	Ref		Ref	
≥1 disease	1.20 (1.05, 1.37)	0.008	1.25 (1.08, 1.45)	0.003
Type of COVID-19 vaccine (*n* = 3783)				
Covaxin	Ref		Ref	
Covishield	1.23 (1.01, 1.49)		1.33 (1.08, 1.64)	
Other	0.17 (0.05, 0.59)	<0.001	0.16 (0.04, 0.58)	<0.001
COVID-19 infection (confirmed)				
Never	Ref		Ref	
Only once	1.21 (0.67, 2.18)		0.98 (0.52, 1.84)	
More than once	1.45 (1.05, 2.01)	0.06	1.20 (0.83, 1.74)	0.61
COVID-19-related oxygen/hospitalization requirement				
No	Ref		Ref	
Yes ^b^	1.20 (0.72, 1.98)	0.46	0.91 (0.52, 1.60)	0.75
Year of last COVID-19 vaccine dose (*n* = 3683)				
2021	Ref		Ref	
2022	3.09 (2.65, 3.58)	<0.001	3.27 (2.80, 3.81)	<0.001

Legend: ^a^ Per-capita income (PCI) in rupees. ^b^ Includes both me and my family member. Model goodness of fit *p*-value: 0.182. “Ref” indicates the Reference category where odds ratio = 1. OR, odds ratio; aOR, adjusted odds ratio; CI, confidence interval. Considering statistical significance at *p* < 0.05.

**Table 3 vaccines-11-01177-t003:** Predictors of willingness to pay for COVID-19 booster dose among non-boosted participants.

Variables	N (col%) Willing to Pay (*n* = 321)	N (col%) Unwilling to Pay (*n* = 1114)	Crude OR (95% CI)	*p*-Value	aOR (95% CI)	*p*-Value
Age (in years)						
50–65	177 (55.14)	671 (60.23)	Ref		-	
>65	144 (44.86)	443 (39.77)	0.81 (0.63, 1.04)	0.10		
Sex						
Female	162 (50.47)	542 (48.65)	Ref		-	
Male	159 (49.53)	572 (51.35)	1.07 (0.84, 1.37)	0.56		
Education						
Illiterate	84 (26.17)	408 (36.62)	Ref		Ref	
Primary	59 (18.38)	221 (19.84)	0.77 (0.53, 1.12)		0.68 (0.46, 1.02)	
Secondary/High school	86 (26.79)	263 (23.61)	0.63 (0.45, 0.88)		0.59 (0.41, 0.85)	
Inter certificate/Graduate and above	92 (28.66)	222 (19.93)	0.49 (0.35, 0.69)	0.01	0.49 (0.33, 0.72)	0.003
PCI ^a^ (in Rupees)						
≤4000	121 (37.69)	573 (51.44)	Ref		Ref	
>4000	200 (62.31)	541 (48.56)	0.57 (0.44, 0.74)	0.01	0.71 (0.53, 0.95)	0.02
Healthcare worker						
No	320 (99.69)	1097 (98.47)	Ref		-	
Yes	1 (0.31)	17 (1.53)	4.96 (0.65, 37.40)	0.12		
Comorbidities						
None	205 (64.06)	804 (72.24)	Ref		Ref	
≥1 disease	115 (35.94)	309 (27.76)	0.98 (0.45, 2.14)	0.005	0.73 (0.55, 0.97)	0.03
Type of COVID-19 vaccine (*n* = 1392)						
Covaxin	27 (9.31)	163 (14.79)	Ref		Ref	
Covishield	258 (88.97)	931 (84.48)	0.59 (0.39, 0.92)		0.56 (0.36, 0.87)	
Other	5 (1.72)	8 (0.73)	0.26 (0.08, 0.87)	0.02	0.35 (0.10, 1.18)	0.024
COVID-19 infection						
Never	307 (95.64)	1063 (95.42)	Ref		-	
Only once	5 (1.56)	13 (1.17)	0.75 (0.26, 2.12)			
More than once	9 (2.80)	38 (3.41)	1.22 (0.58, 2.55)	0.78		
COVID-19-related oxygen/hospitalization requirement						
No	315 (98.13)	1097 (98.47)	Ref		-	
Yes ^b^	6 (1.87)	17 (1.53)	0.81 (0.31, 2.08)	0.67		
Year of last COVID-19 vaccine dose (*n* = 1392)						
2021	149 (51.38)	422 (38.29)	Ref		Ref	
2022	141 (48.62)	680 (61.71)	1.70 (1.31, 2.21)	<0.001	1.52 (1.16, 1.98)	0.002

Legend: ^a^ Per-capita income (PCI) in rupees. ^b^ Includes both me and my family member. Model goodness of fit *p*-value = 0.193. “Ref” indicates the Reference category where odds ratio = 1. col% indicates the column-wise percentage. OR, odds ratio; aOR, adjusted odds ratio; CI, confidence interval. Considering statistical significance at *p* < 0.05.

## Data Availability

The data that support the findings of this study are available from the corresponding author upon reasonable request.

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
