# Peer review of "COVID-19 Booster Dose Coverage and Hesitancy among Older Adults in an Urban Slum and Resettlement Colony in Delhi, India"

_vaccines, 2023, doi:10.3390/vaccines11071177_

Round 1

Reviewer 1 Report

The title is too long. Please shorten it.

May be “A cross sectional study on the covid-19 booster dose in elderly

in urban slum and resettlement colony of New Delhi” or can be edited

Abstract is not in the style of the journal but written in a good manner.

Keywords:  Booster dose; Booster dose hesitancy are almost same

Introduction: In general this section is too long than expected.

The entire sections can be divided into three para, 1) About the need of Vaccine and current status in Delhi or India, 2) About the hesitancy nature and literature review, 3) Gap and about the hypothesis or need of the study.

So many paragraphs lost the test of this section.

Line 39: Please update the data

Line 36-40: Please write something about non-vaccine approaches also as mentioned in https://pubmed.ncbi.nlm.nih.gov/34162043/,   https://pubmed.ncbi.nlm.nih.gov/35137366/ and . https://pubmed.ncbi.nlm.nih.gov/32339832/ and these works were done in special reference to India. They may be incorporated.

Line 45: At some time point WHO also suggested to not take booster dose. Please mention that also.

Line 54-56: Mention about their approval by WHO.

Line 67” Why abbreviated again? Please follow this rule though our the ms.

Line 124 Materials methods

Please include a consort flow for the study

Line 133: What was the drive to include only this age group?

149-152: May be a questionnaire? No?

Line 168-176: Move this part into the beginning of this section.

Line 161-166: Divide it into a section as Statistical analyses

Results

This section is ok

Discussion:

This section I find mostly the repetitions of the results at many points. So, please discuss the results that why hesitation arrives among people and what could be the possible way to make them able to understand the fact about the booster dose.

This is very important that a model must be presented in the study citing the results (problem of hesitancy) and its solution by Govt, society or others. It should be a pictorial presentation.

The language is ok for this ms. 

Reviewer 2 Report

The introduction section needs to be improved. The authors have jumbled up the information that makes the introduction part unorganized.

The sample size is low. It is not very understandable how analyzing the subjects of just one particular area would help. The authors should also cover other regions and then a comprehensive analysis should be made.

The authors should clearly mention  how the information for each subject was obtained for the study. As the subjects are the old people or illiterate, how did the authors ensure the authenticity of the information? What do they mean when they mention "field investigators also attempted to verify the Covid-19 vaccination status using the paper or electronic vaccine certification record." Was that verified for every individual?

The authors need to stress how this study is important and impactful.

Overall the style of writing needs to be improved to make the study  understandable.

Round 2

Reviewer 2 Report

I would suggest major revision is required for this manuscript. I have the following comments/suggestions: The authors have improved the introduction section and have also added a schematic as Figure 1 which makes the manuscript reader friendly. However, I still have my concerns about the generalizability and how the obtained data would be useful in the broader context. The authors have added a few lines about the importance of the study. It's not very understandable and convincing. If they could elaborate a little more how the data would be used in planning the vaccination strategy in that region or in the execution, the study would appear more relevant.

none

Round 3

Reviewer 2 Report

The authors have tried addressing the comments from the review. Still the overall study design and its application does not look convincing to me. 

The authors mention:

" In India wherein an estimated 49% of the population lives in urban slums/resettlement colonies characterized....."

Are they talking about their specific study group or the population in general?

How the data from one urban slum would be helpful?

The authors mention that such studies would help identify high risk population and awareness can be spread among these vulnerable population. Do authors have data that subject groups staying in non urban slums-setup are more aware and open towards vaccination?

It is not advised to draw such conclusions without having a control group.

The authors shuld consider generating data from other set ups and also from other locations. Such comapative analysis can help in planning and execution of vaccine programs and make the study relevant.

The quality of English needs to be improved.

Author Response

Thank you for your comments. Our response is provided below:

  1. Still the overall study design and its application does not look convincing to me. 

The cross-sectional design is the most appropriate for achieving the study objective which was “to estimate Covid-19 vaccination including precaution (booster) dose coverage”. Our objective was never to conduct a comparative analysis of coverage in different areas such as slums and non-slums. The justification for the study was that booster dose coverage rates in urban slum settings from India were unavailable.

  1. Are they talking about their specific study group or the population in general? How the data from one urban slum would be helpful?

We would like to state that of the entire Indian urban population: nearly half (49%) lives in urban slums which have common adverse social determinants including more difficult access to healthcare and lower SES (Reference 36 and 37). Therefore  our study findings have considerable generalizability when extended to other socioeconomically disadvantaged populations in India..

  1. Do authors have data that subject groups staying in non urban slums-setup are more aware and open towards vaccination? It is not advised to draw such conclusions without having a control group

In our study, during multivariable analysis, we found that higher educational status and higher SES, were independent predictors of Covid-19 booster dose vaccine acceptance. As with cross-sectional surveys, the comparison group was deduced as those participants who did not have the outcome (i.e. unvaccinated/did not receive booster) of interest. This is because not all people living in urban slums are socioeconomically poor / neither do all lack education, so we have a comparative group (higher SES, higher educational status) within our study population  it-self.

  1. The authors should consider generating data from other set ups and also from other locations. Such comapative analysis can help in planning and execution of vaccine programs and make the study relevant

We appreciate the suggestion to do a multicentric study but we currently lack the time and resources for the same. Consequently, we acknowledge this limitation that may restrict the generalizability of our study findings. Nevertheless, most urban slums share common adverse social determinants which to an extent improves the external validity of our findings, at-least in Delhi, a state with 20 million population.